# Layer-dependent exciton polarizability and the brightening of dark excitons in few-layer black phosphorus

Yuchen Lei[1], Junwei Ma[1], Jiaming Luo[1], Shenyang Huang [1], Boyang Yu[1], Chaoyu Song [1], Qiaoxia Xing[1], Fanjie Wang[1], Yuangang Xie[1], Jiasheng Zhang[1], Lei Mu[1], Yixuan Ma[1], Chong Wang[2,3] & Hugen Yan [1] ✉

The evolution of excitons from 2D to 3D is of great importance in photophysics, yet the layer-dependent exciton polarizability hasn't been investigated in 2D semiconductors. Here, we determine the exciton polarizabilities for 3- to 11-layer black phosphorus—a direct bandgap semiconductor regardless of the thickness—through frequency-resolved photocurrent measurements on dual-gate devices and unveil the carrier screening effect in relatively thicker samples. By taking advantage of the broadband photocurrent spectra, we are also able to reveal the exciton response for higher-index subbands under the gate electrical field. Surprisingly, dark excitons are brightened with intensity even stronger than the allowed transitions above certain electrical field. Our study not only sheds light on the exciton evolution with sample thickness, but also paves a way for optoelectronic applications of few-layer BP in modulators, tunable photodetectors, emitters and lasers.

Few-layer black phosphorus (BP) is always a direct bandgap semiconductor, regardless of the thickness[1–11]. This renders an ideal platform to interrogate the exciton property evolution from quasi-2D to 3D[9,12], such as the exciton binding energy[13] and the oscillator strength[11]. Such practice has been hindered in the popular layered transition metal dichalcogenides, such as $MoS_2$ and $WSe_2$, due to the fact that they possess direct bandgap only in the monolayer form[14,15]. As another important parameter, the exciton polarizability[16,17] of few-layer BP in the out-of-plane direction is responsible for the field-induced dipole moment and governs the quantum confined Stark effect (QCSE)[18,19], manifested by the exciton energy shift under a vertical electrical field. Its exact value and layer dependence are crucial for electro-optical effect and various optoelectronic applications. However, experimental determination of the exciton polarizability is still lacking, even though a widely tunable electrical gap with a vertical electrical field has been demonstrated in few-layer BPs through the transport technique[20–24]. Optical absorption and scanning tunneling microscopy (STM) studies of few-layer BP controlled by a single gate have revealed the shift of the exciton resonance or the quasiparticle gap[25–28]. Nevertheless, such studies struggle to differentiate the electrical field effect and doping-induced band-filling effect[29], therefore the determination of the exciton polarizability could not be achieved.

Here, through the broadband photocurrent spectroscopy[30–32] of dual-gate few-layer BP devices, we systematically study the pure field-induced shifts of the optical bandgaps for 3- to 11-layer BP and determine the exciton polarizability. We reveal that a simple quantum well model[18,19,33,34] works well for the thin samples but fails in the relatively thick ones, which requires consideration of free carrier screening[35–38]. Meanwhile, we uncover the behavior of higher-index excitons under the field, which was unattainable in previous transport studies. Most intriguingly, dark excitons are brightened with the symmetry-breaking field[26,39], which promises applications in light modulation[5,40] with large modulation depth.

[1]State Key Laboratory of Surface Physics, Key Laboratory of Micro and Nano-Photonic Structures (Ministry of Education), and Department of Physics, Fudan University, Shanghai 200433, China. [2]Centre for Quantum Physics, Key Laboratory of Advanced Optoelectronic Quantum Architecture and Measurement (MOE), School of Physics, Beijing Institute of Technology, Beijing 100081, China. [3]Beijing Key Lab of Nanophotonics & Ultrafine Optoelectronic Systems, School of Physics, Beijing Institute of Technology, Beijing 100081, China. ✉e-mail: Hgyan@fudan.edu.cn

## Results

### Device fabrication and characterization

The schematic and optical image of a typical BP photodetector based on a dual-gate transistor structure are shown in Fig. 1a, b, respectively. Here the BP film is in contact with graphite flakes, which serve as the source and drain electrodes. Sandwiched by hBN flakes, the BP flake was transferred onto a piece of graphite using the polypropylene carbonate (PPC) hot pick-up technique[41]. Subsequently, another thin graphite flake was transferred on top. All exfoliation and transfer processes were performed in a nitrogen-filled glovebox with oxygen and moisture concentrations below 0.1 part per million (ppm) to prevent BP from degradation. Afterwards, chromium-gold electrodes were deposited. Source/drain graphite flakes were arranged along the armchair direction to maximize the photocurrent[5,20,42] (for more details, see Methods). The quality of such hBN/BP/hBN devices can be maintained for a relatively long period without noticeable degradation[43–46].

Photocurrent spectra were obtained by a Fourier transform infrared spectrometer (FTIR) operated in the step-scan mode (See Methods and Supplementary Fig. 1). Figure 1c shows a typical normalized photocurrent spectrum of an ungated 7-layer BP sample. Detailed normalization process is presented in Supplementary Fig. 3 and Note 1. Two clear peaks can be identified from Fig. 1c, representing excitons associated with $E_{11}$ and $E_{22}$ transitions[9]. Those two peaks, with energies in good agreement with the absorption spectrum (Supplementary Fig. 2), are originated from transitions from valence subbands to conduction subbands, as shown in the inset of Fig. 1c. Note that the photocurrent spectrum is equivalent to the absorption spectrum, but it's more convenient to measure—particularly for such dual-gate devices with multiple stacked 2D materials—due to the less stringent requirement than that for the absorption measurement in the infrared range, for which, large sample area and simple sample environments are necessary[9].

### Quantum confined Stark effect of $E_{11}$ excitons

Before detailed photocurrent studies of the Stark effect[19], we first performed four-terminal transport measurements to examine the gating performance of the device. Figure 2a shows the conductance as a function of the back gate voltage ($V_{bg}$) in a 6-layer BP device at different static top gate bias ($V_{tg}$) ranging from −6.0 V to 8.0 V at room temperature. In the dual-gate configuration, two electric displacement fields, $D_t = \varepsilon_0 \varepsilon_b (V_{tg} - V_{t0})/d_t$ and $D_b = \varepsilon_0 \varepsilon_b (V_{bg} - V_{b0})/d_b$ are applied in top and back gate dielectrics to control the doping and electrical field across the thin BP film, where $\varepsilon_b = 3.1$ is the relative permittivity of boron nitride[20], $d_t$ and $d_b$ are thicknesses of the top and bottom hBN respectively, and $V_{t0}$ and $V_{b0}$ are charge-neutral point voltages due to the unintentional doping respectively. When we decrease $V_{tg}$, the position of charge neutral point shifts to higher $V_{bg}$ linearly (Supplementary Fig. 4), with a slope equal to the thickness ratio of the top and back hBN, indicating that the free carriers due to $V_{tg}$ are fully compensated. Recently a few studies determined the change of the electrical gap of BP by transfer curves under certain field[20–22,24]. Our transport results are fully consistent here.

Next, let us examine the frequency-resolved photocurrent and its field dependence. By measuring at charge neutral points determined from its transfer curve, we eliminate Burstein Moss Shift (BMS)[47–50] caused by Pauli blocking of the optical transitions, rendering an environment with pure vertical electrical field and a fixed Fermi level within the bandgap, as illustrated in Fig. 2b. The out-of-plane electric field leads to band-bending across the quantum confined system and alters the conduction and valence subband energies, resulting in the QCSE. Figure 2c shows normalized photocurrent spectra of a 6-layer BP sample at room temperature with different displacement fields. The resonance associated with the optical bandgap is the major feature of the photocurrent spectrum, consistent with former absorption studies[9,11,13]. The field we use as our x-axis is $D/\varepsilon_0$ (the continuity condition gives us $D_b = D_t = D_{BP} = D$, with $D_{BP}$ as that inside BP). As the electrical field is applied, the exciton resonant energy hardly changes at low fields and then at higher fields, shifts to lower energy with an increasing rate until the peak becomes too weak under relatively large fields. The $E_{11}$ peak can be continuously tuned from 0.552 eV at zero field to 0.509 eV under a moderate external field of 1.07 V nm⁻¹, demonstrating the tunable optical gap in the mid-infrared range. The photocurrent spectra for the reversed field from 0 to −0.75 V nm⁻¹ are shown in Supplementary Fig. 5, exhibiting almost identical response. Figure 2d displays the extracted peak position as a function of the applied field. With the quantum mechanical perturbation theory and simple symmetry considerations, it has been shown that for symmetric quantum wells, the linear dependence of the energy shift on the electric field vanishes, leaving a quadratic dependence[18,19,33,34],

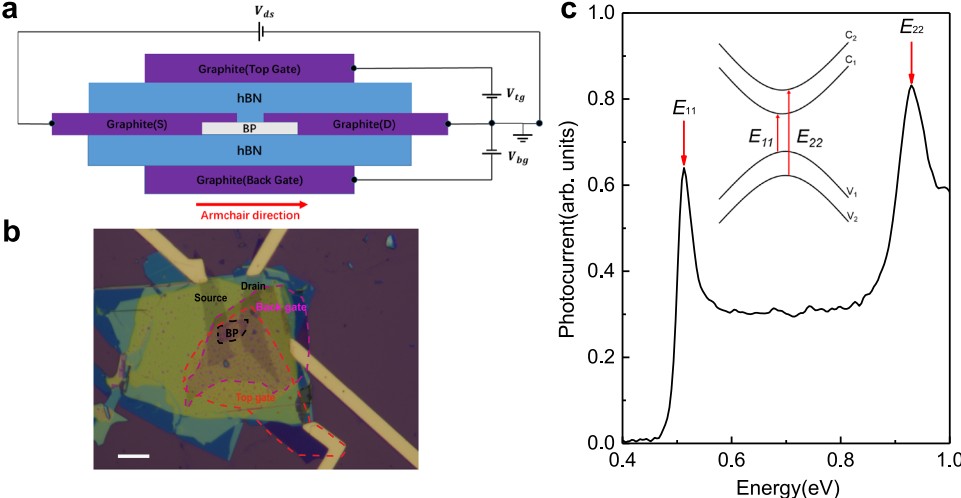

**Fig. 1 | Structure of the BP mid-IR photodetector based on a hBN-encapsulated dual-gate configuration. a** Schematic of dual-gate hBN/BP/hBN field effect transistor (FET). **b** Optical images of a dual-gate BP FET. Red and violet frames represent the region of top and bottom graphite gates. To make sure the electrical field is uniform, the BP flake is fully covered by top and bottom hBN and graphite flakes. The scale bar is 20 μm. **c** The normalized photocurrent spectrum of an intrinsic 7-layer BP. The major features (indicated by red arrows) stand for excitons from $E_{11}$ and $E_{22}$ transitions. Top inset is a schematic illustration for optical transitions between different subbands.

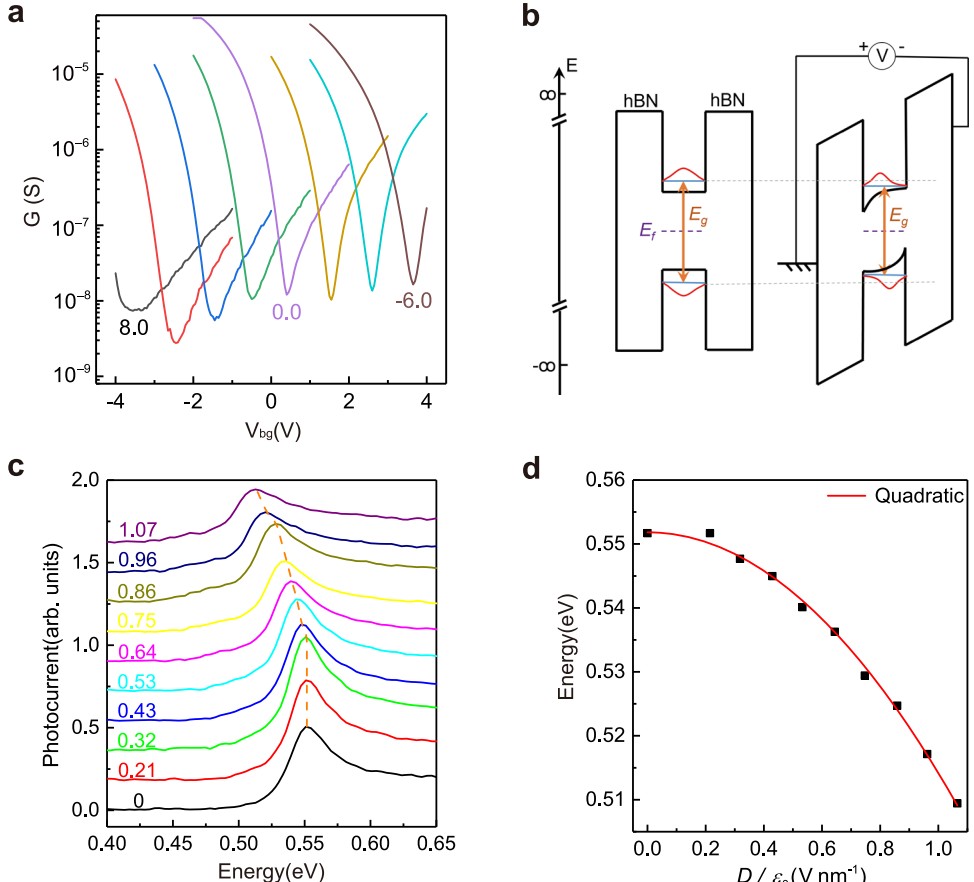

**Fig. 2 | Field effects on the transport and photocurrent response of a 6-layer BP device. a** The conductance of the dual-gate BP transistor as a function of the bottom gate ($V_{bg}$) at different top gate biases ($V_{tg}$) from −6.0 V to 8.0 V. The source-drain bias is 200 mV. **b** Schematic energy band diagram, and wave functions of the ground states of an intrinsic BP QW and the same QW under an electric field, illustrating QCSE. **c** The normalized photocurrent spectra of the 6-layer BP device under a series of electrical field from 0 V nm$^{-1}$ to 1.07 V nm$^{-1}$. The dashed line indicates the evolution of the peak position. Spectra are shifted vertically for clarity. **d** The $E_{11}$ position (black dot) extracted from Fig. 2c as a function of external field. The red solid curve represents a quadratic fit of the experimental data.

with the coefficient related to the exciton polarizability[16,17,25]. In other words, due to the symmetry of the quantum well, the unperturbed exciton lacks a dipole moment along z-direction (out-of-plane). We also carried out the fitting of the 6-layer Stark shift in Fig. 2d with a quadratic dependence: $E = E_0 - \alpha \cdot (D/\varepsilon_0)^2$, where the fitting parameter $\alpha = 0.048$ eV nm$^2$ V$^{-2}$ is the exciton polarizability, and $E_0 = 0.552$ eV is the gap at zero field. The nice quadratic fitting in the figure suggests zero original exciton dipole moment along z-direction, which otherwise contributes a linear field dependence to the exciton energy shift. Therefore, through the quadratic fitting of the Stark shift, we obtained the exciton polarizability. The same scheme will be applied to other thickness samples.

**Layer-dependent QCSE of $E_{11}$ excitons**

Now we interrogate the QCSE of $E_{11}$ in BP samples of different thickness. For instance, as we increase the electrical field to 1.50 V nm$^{-1}$, $E_{11}$ peak of a 3-layer sample shifts from 0.843 eV to 0.827 eV, dropping only 0.016 eV (Supplementary Fig. 6). The shift is apparently smaller than that of the 6-layer sample described in Fig. 2d, which shifts 0.022 eV under a much smaller field of 0.64 V nm$^{-1}$, indicating the relevance of the sample thickness. We systematically studied the QCSE for 3- to 11-layer samples and the extracted $E_{11}$ peak shifts of representative 3-, 6- and 10-layer BP samples are shown in Fig. 3a, revealing more prominent shifts for thicker samples. The reproducibility was checked, as shown in Supplementary Fig. 7.

As we did for the 6-layer BP in Fig. 2d, we extracted the exciton polarizability α of $E_{11}$ peaks for all 3- to 11-layer BP samples, as shown in Fig. 3b. The extraction process involving fittings is detailed in Supplementary Fig. 8. It's clear that as the layer number increases, the polarizability saliently increases. This behavior is qualitatively consistent with the predicted thickness dependence through a simple perturbation method, which gives an $L^4$ scaling for the polarizability[18,34] ($L$ is the thickness or layer number). However, our result doesn't show such dramatic dependence. We tried to fit the experimental polarizability α against layer number $L$ (proportional to the well width) with a phenomenological polynomial: $\alpha = A + B \cdot L^C$. The result is shown in Supplementary Fig. 9 with the power $C = 0.502$. This suggests the inadequacy of the simple perturbation method.

To apprehend the thickness dependence of the QCSE more properly, we performed exact calculations based on an ideal one-dimensional infinite Quantum Well (QW) under an electric field[18,19,33,34]. The eigenstates were obtained analytically with the wave functions expressed as linear combinations of two independent Airy functions (see Supplementary Note 2). By neglecting the field-induced change of the exciton binding energy, shifts of exciton resonant energy under different electric fields were obtained by summing the shifts of valence and conduction subbands, as shown in Supplementary Fig. 10 (3-layer and 6-layer results are also shown in Fig. 3a as solid curves to compare with experimental data). We can conclude from the calculation that the shift is enhanced with the growing external field as well as the widening of the QW, which is consistent with the trend of our experimental data.

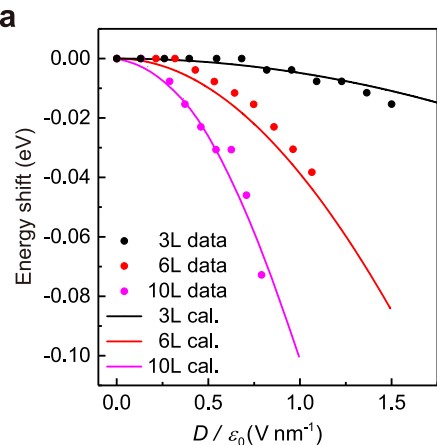
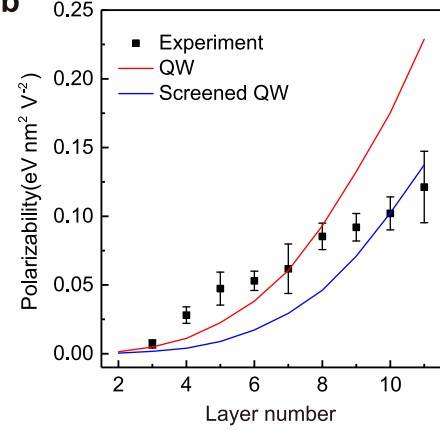

**Fig. 3 | The layer dependent QCSE. a** The $E_{11}$ exciton energy shifts of 3, 6 and 10 L BP devices. The dots represent the experimental data while the curves depict the QW calculation results for the 3 and 6 L, and the screened QW model calculation for the 10 L, respectively. **b** The experimental and theoretical polarizabilities ($\alpha$) of 3- to 11-layer BP devices acquired by fitting the experimental data and calculations

respectively. The QW- and screened QW-based results are shown by the red and blue curves respectively. The error bars of the experimental data points are included, which originate from the measurements of the spectrum and the thickness of BN dielectrics.

However, there is quantitative disagreement for relatively thick samples. Supplementary Fig. 11 shows the calculated results and the data of the 10-layer sample. We can see that the calculation overestimates the Stark shift. This can be attributed to the arising of the electrostatic screening effect in experiment, possibly due to the existence of thermal carriers, given that the bandgap is relatively small in few-layer BP and further shrinks with increasing layer number. Such screening makes the local field inside BP smaller. The signature of the screening is the non-linear shift of charge neutral points in the transfer curve. Previous transport studies have shown that the screening could occur in 10 nm BP samples[21]. Although our transport measurements could also affirm the screening effect in BP with similar thickness, as shown in Supplementary Fig. 12 for comparison of BP samples with 13- and 20-layer (about 10 nm), it may exist in thinner BP samples, but may not be manifested saliently in the transfer curve.

In order to take into account the screening effect in BP films under different electric fields, we resorted to the nonlinear Thomas-Fermi theory, which has been applied to films of various materials to calculate the charge distribution and screening[35–38]. We also performed calculations of the bandgap dependent carrier density resulted from thermal excitation to verify the feasibility of the Thomas-Fermi theory. It turns out that for BP samples thicker than 9 layers (about 4.5 nm), thermally excited free carriers are sufficient and the screening needs to be considered (see Supplementary Note 4). By still assuming a uniform electric field inside the BP film, we recalculated the QW model under a screened local electrical field. The resulted QCSE of 10-layer BP is shown by solid line in Fig. 3a. The screened QW model agrees better with experimental results than the QW model for relatively thick samples. This clearly indicates the necessity for the appropriate electrostatic screening correction.

Based on the calculated $E_{11}$ energy versus field for 2- to 11-layer devices with both models, now we can extract the theoretical exciton polarizability $\alpha$ by fitting the results with a quadratic form, as shown in Fig. 3b. It's apparent that the QW model gives better description for samples below 8-layer, while the screened QW model works better for thicker ones. Admittedly, neither model can fit the overall data well enough, and more sophisticated theoretical work is required in the future.

## Index-dependent QCSE and the brightening of dark excitons

Apart from the resonance associated with the optical bandgap, more inclusively, we could gain insights into higher-index exciton transitions

along the way, which were hardly attainable in previous photoluminescence[51] and transport[20–22,24] studies. Typically, devices at cryogenic temperature exhibit better signal-to-noise ratio in the photocurrent spectroscopy, so that subtle features can be resolved. We measured the photocurrent spectra of a 10-layer sample at 77.5 K. Figure 4a displays the field dependent photocurrent spectra. Two main peaks can be clearly observed in the ungated spectrum (black line) at 0.433 eV and 0.697 eV, representing the $E_{11}$ and $E_{22}$ transitions[9]. $E_{11}$ and $E_{22}$ peak positions at different fields were extracted and plotted in Fig. 4b by a series of blue and orange triangles. As the electrical field increases, $E_{11}$ peak shifts to lower energy region, revealing robust QCSE, while $E_{22}$ peak shifts much more slowly. Similar scenario was observed in all 7-layer to 11-layer BP samples, whose $E_{22}$ transitions are within the measurement range, such as the 7-layer sample shown in Supplementary Fig. 13. We carried out calculations for the 10-layer based on the screened QW model[33–38] and the results are consistent. Supplementary Fig. 14 shows the calculated evolution of the first ($l = 1$) and the second ($l = 2$) valence and conduction subbands. We can see that $l = 1$ subbands shift much quicker than $l = 2$ subbands, indicating index-dependent QCSE. Typically, the ground state wave function is more sensitive to external perturbations, such as the tilting of the QW potential induced by an vertical field[34], than its excited state counterparts. Hence, $E_{11}$ transitions show stronger QCSE. More detailed discussions are presented in Supplementary Note 3.

In addition to main peaks, some new features emerge between them when the field is applied[39]. Notably, two new peaks arise between $E_{11}$ and $E_{22}$ (as indicated in Fig. 4a red frame). The scenario fits well with the brightening of forbidden transitions (dark excitons) and as a result, the peaks could be assigned to forbidden transitions $E_{21}$ and $E_{12}$ (see the inset of Fig. 4b). As we know, for a symmetric QW, the dipole selection rule doesn't allow optical transitions with odd $\Delta l$ ($\Delta l$ is the difference between the valence and conduction subband indices). Only with symmetry breaking, such transitions occur[52]. With a vertical gate electrical field, the well is not symmetrical anymore, as sketched in Fig. 2b, resulting in the appearance of dark excitons. Previously, such new peaks, though with weak amplitude, were observed in unintentionally doped few-layer BP in an unpredictable manner[9]. Importantly, now we can readily brighten them with a gate field. Owing to the optimized signal to noise ratio, here we can distinguish them clearly and extract the peak positions. Because of the difference in the effective masses along the out-of-plane direction for the electron and hole, the energies of $E_{21}$ and $E_{12}$ are slightly different. Previous

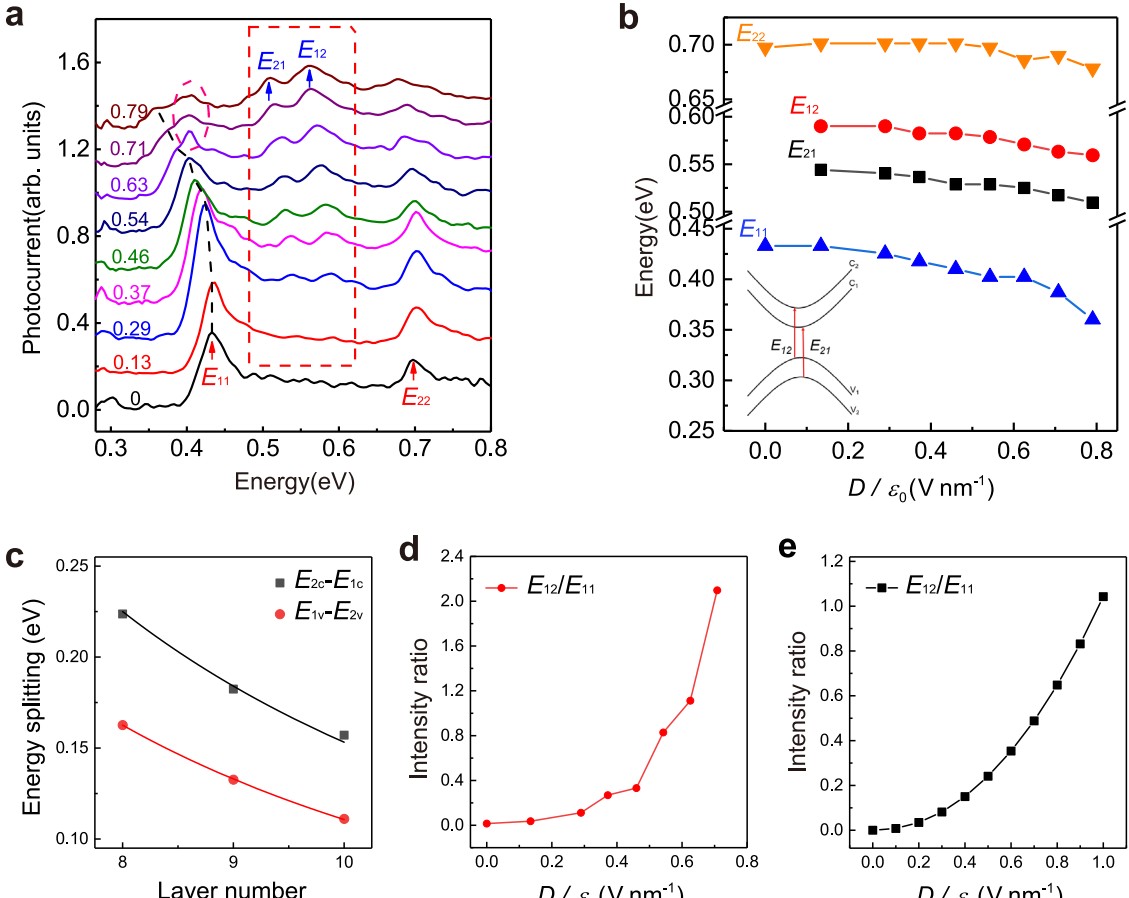

**Fig. 4 | Higher-index transition and brightening of dark excitons. a** The normalized photocurrent spectra of a 10-layer BP device measured at 77.5 K under a series of external fields. The unit of the numerical labels for the field in the figure is V nm$^{-1}$. The red dashed frame highlights the emerging $E_{21}$ and $E_{12}$ transitions. The peak marked by a magenta dashed circle is an artifact from ice. The black dashed line indicates the evolution of the $E_{11}$ peak. Spectra are vertically shifted for clarity. **b** The peak positions extracted from Fig. 4a as a function of external field. The inset is a schematic illustration for the forbidden optical transitions $E_{21}$ and $E_{12}$. **c** The conduction and valence band splittings extracted from our 8- to 10-layer photocurrent spectra shown in Supplementary Fig. 18, Supplementary Fig. 19 and Fig. 4a respectively. The black and red curves are the fitting results based on Eqs. (2) and (3) respectively. **d** The intensity ratio of $E_{12}$ to $E_{11}$ transitions as a function of field extracted from Fig. 4a. The intensities were extracted by Lorentz fitting of the peaks in the spectra. **e** Numerical calculation of the corresponding intensity ratio.

experiment[53] has shown that for BP films, the splitting of conduction bands is larger than that of valence bands, so $E_{21}$ has slightly lower energy, as assigned in Fig. 4a. Supplementary Fig. 15 shows photocurrent spectra of the same 10-layer BP device at room temperature. Although the signal to noise ratio is not ideal due to the thermal noise, two bumps can still be identified when field increases. Similar scenario also occurs in other samples, such as the 8-layer BP shown in Supplementary Fig. 16.

As expected, $E_{21}$ and $E_{12}$ transitions also shift with electrical field. We picked up their peak positions from where the peaks are resolvable (field above 0.17 V nm$^{-1}$ experimentally) and display them in Fig. 4b. Both of the transitions shift to lower energy as the field increases, and the paces are nearly the same. The shifts are largely consistent with our calculations based on the screened QW model, as shown in Supplementary Fig. 17. As discussed above, $l = 2$ subbands are inert to the field. Consequently, the shift of those two forbidden transitions can be attributed to the $l = 1$ bands, i.e., $E_{21}$ is dominated by the first conduction band $c_1$ and $E_{12}$ by the first valence band $v_1$. From this point of view, the nearly identical shift pace of forbidden transitions suggests similar shift rates of bands $c_1$ and $v_1$, consistent with calculations as well (Supplementary Fig. 14).

The accurate extraction of the energies of the dark excitons can help us gain deeper insights into the electronic structure of

few-layer BP. The energy difference of $E_{12}$ and $E_{11}$ peaks give us the splitting of the $l = 1$ and $l = 2$ conduction bands, while the gap between the $l = 1$ and $l = 2$ valence bands can be measured by the difference of $E_{21}$ and $E_{11}$ peaks. Such respective splittings cannot be obtained experimentally without resorting to forbidden transitions. By neglecting the shift of bands under relatively small electric field, the original band splittings of 8- to 10-layer BP system are extracted and shown in Fig. 4c (the spectrum and the peak position extraction of the 8-layer and 9-layer BP are displayed in Supplementary Fig. 18 and Supplementary Fig. 19). It is apparent that the splitting of conduction bands is larger than that of valence bands, and the $l = 1$ and $l = 2$ bands move closer as the thickness grows. The shifting trend of band splitting has been calculated in previous studies[9,53] with a phenomenological tight binding model, in which the eigenvalues of a N-layer BP system at Brillouin zone center can be described as:

$$E_{nj} = E_{1j} - 2\gamma_j \cos\left(\frac{n\pi}{N+1}\right), \tag{1}$$

where $\gamma$ stands for the nearest neighbor interlayer interaction, $j = c, v$ represents the conduction or valence band, $n = 1, 2, \cdots, N$ and $E_{1j}$ is the band energy of monolayer BP. The difference of $E_{1c}$ and $E_{2c}$ can be

calculated by Eq. (1) as:

$$E_{2c} - E_{1c} = -2\gamma_c \left[ \cos\left(\frac{2\pi}{N+1}\right) - \cos\left(\frac{\pi}{N+1}\right) \right], \qquad (2)$$

and the same can be applied to valence band as:

$$E_{2v} - E_{1v} = -2\gamma_v \left[ \cos\left(\frac{2\pi}{N+1}\right) - \cos\left(\frac{\pi}{N+1}\right) \right]. \qquad (3)$$

It's reasonable to assume that $\gamma_c$ and $\gamma_v$ are constants regardless of thickness, the band splitting estimated by Eqs. (2) and (3) both shrink as the layer number $N$ becomes larger, perfectly consistent with our data shown in Fig. 3a. Fitting the experimental results by Eqs. (2) and (3) can give us the actual value of the interlayer couplings, that is $\gamma_c = 0.648$ eV and $\gamma_v = -0.468$ eV, consistent with previous results calculated from another series of forbidden transitions[53].

As the field increases, the forbidden transitions become more and more prominent, as shown in Fig. 4a, indicating that the external field brightens and strengthens these transitions. Surprisingly, the amplitude of the $E_{12}$ transition even becomes stronger than the originally allowed $E_{11}$ and $E_{22}$ transitions with field exceeding 0.6 V nm$^{-1}$. The brightening is fully reversible and readily controllable by the gate field. When an external electrical field is applied, electrons and holes are dragged towards opposite walls of the well. As a result, the electron and hole envelope wave functions are no longer sinusoidal, and all the overlap integrals are in general non-zero, brightening the otherwise forbidden transitions. Particularly, the $\Delta l = 1$ forbidden transitions associated with $c_1$ or $v_1$ bands appear to be the most sensitive to the field. To gain insight into the brightening process, we took the intensity ratio between the forbidden transition $E_{12}$ and the allowed transition $E_{11}$ to illustrate the enhancement of forbidden transitions under increasing field, given that $E_{12}$ is stronger than $E_{21}$ and hence provides a more accurate measure. Figure 4d shows the intensity ratio extracted from spectra in Fig. 4a by Lorentz fittings. It is apparent that the ratio tends to increase with an accelerating speed as the external field grows. Based on Fermi's golden rule, we estimated the intensity ratio by computing the overlapping integrals of the envelope wave functions under a vertical field. The numerical calculation result is shown in Fig. 4e. Detailed calculation method and discussions are presented in Supplementary Note 5. As the electrical field increases, the ratio enlarges significantly. Our experimental result has the same trend as the calculated counterpart. However, the absolute values differ, and possible reasons are discussed in Supplementary Note 5. Overall, the deviation is reasonable and qualitatively, our experiment is still quite consistent with our numerical result, unraveling the mechanism for the brightening of dark excitons. The drastic change of the forbidden transition intensity from zero to even stronger than the allowed ones, suggests that it can provide a new mechanism for light intensity modulation, potentially with large modulation depth.

## Discussion

The dual-gate configuration in our study is capable of applying a pure electric field on BP, without changing the Fermi level through doping. This is radically different from previous single gate studies, where Pauli blocking induced Burstein–Moss effects coexists with the QCSE[26,29,39]. The Burstein–Moss effects caused by the electrostatic doping in BP lead to a blue shift of the band edge absorption due to the Pauli blocking of lower transition energies. Therefore, the single gate experiments encounter combinations of competing QCSE and Burstein–Moss effects, leading to nonmonotonic shifts of absorption peaks, as predicted by calculations[29]. As a comparison, we also tried the single gate configuration. Supplementary Fig. 20 displays the photocurrent spectra of a single-gate 13-layer sample with and without gate voltage. We can see a slight blueshift of the $E_{11}$ peak with gate, in

sharp contrast to our dual-gate results, indicating the Burstein–Moss effects overshadow QCSE. This suggests the necessity of dual-gate to truly manifest the QCSE and to extract the exciton polarizability.

The increase of the exciton polarizability is only one of the scenarios when the sample thickness increases. As a matter of fact, another key ingredient in QCSE is the robustness of excitons under an intense electric field due to the quantum confinement of the electrons and holes. This is most vividly manifested in our 3-layer BP device shown in Supplementary Fig. 6, where the exciton still has significant oscillator strength with a field as large as 1.5 V nm$^{-1}$. When the quantum confinement eases, the oscillator strength of the exciton becomes more vulnerable to the field, as seen in Supplementary Fig. 23 for the $E_{11}$ transition of the 20-layer sample. The intensity of the $E_{11}$ exciton already decreases a lot at 0.06 V nm$^{-1}$ and becomes hardly discernable with field above. The scenario is also manifested in 13-layer and 18-layer BP films shown in Supplementary Figs. 21 and Fig. 22. Based on this trend, it is reasonable to infer that for bulk-like BP, a much smaller field can already separate the electron and hole to a large extent, resulting in a diminishing exciton oscillator strength.

In summary, through meticulous photocurrent spectroscopy of high-quality dual-gate BP devices, we obtained the layer-dependent $E_{11}$ exciton polarizability and revealed the carrier screening effect in relatively thicker samples. This provides insights into the dimensional crossover of excitons in layered materials. In addition to the lowest energy excitons, we also examined the higher-index subband excitons and brightened dark excitons. Benefiting from the emergence of the forbidden transitions, we acquired the respective interlayer couplings for the conduction and valence bands. Our work lays the foundation for electro-optics based on BP and underlines the great potential for optoelectronic applications.

## Methods

### Fabrication of hBN-sandwiched BP photodetectors

BP crystals were purchased from HQ Graphene with purity >99.995%[54,55]. BP thin flakes were first mechanically exfoliated from bulk crystals onto low viscous polydimethylsiloxane (PDMS) substrates in a nitrogen-filled glovebox with oxygen and moisture concentrations below 0.1 part per million (ppm). Before transferring, polarization-dependent extinction spectra were measured to characterize the thickness and lattice direction of BP. Shortly afterwards, the BP flake was assembled together with hBN flakes (thickness measured by a DektakXT profilometer) and graphite electrodes onto silicon substrates covered with a 285-nm thick SiO$_2$ using the polypropylene carbonate (PPC) hot pick-up technique described in ref. 41. Then the AZ-5214 photoresist was spun onto the samples and a Direct Writer (uPG501) was used to define the shape of multiple electrodes. The exposed top hBN layer was etched through the reactive ion beam etching (Trion T2) in a SF$_6$ (50 standard cubic centimeter per minute) environment. Finally, chromium/gold (5/85 nm) films were evaporated through thermal evaporation (Nano36) to form contacts.

### Transport and photocurrent measurements

The dual-gate transport measurements were performed using a semiconductor characterization system (Keithley 4200). The photocurrent measurements were then carried out by a FTIR (Bruker Vertex 70 v) in conjunction with a Hyperion 2000 microscope, operated in the step-scan mode. A lock-in amplifier (SR830) was employed to obtain the photocurrent signal by modulating the light source of a tungsten halogen lamp with a chopper. The lock-in output was fed back to FTIR as the input. The interferogram obtained in this way was Fourier-transformed by the FTIR software to display spectra in the frequency domain. The low-temperature measurements were carried out in a liquid nitrogen cryostat (Janis Research ST-300) with a pressure of about $1 \times 10^{-6}$ mbar.

## Data availability

All relevant experimental data are presented in the paper and the Supplementary Information. The data that support the findings of this study are available from the corresponding author upon request.

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

## Acknowledgements

H.Y. is grateful to the financial support from the National Key Research and Development Program of China (Grant Nos. 2022YFA1404700,2021YFA1400100), the National Natural Science Foundation of China (Grant No. 12074085), the Natural Science Foundation of Shanghai (Grant No.23XD1400200). S.H. acknowledges the financial support from the China Postdoctoral Science Foundation (Grant No. 2020TQ0078). C.W. is grateful to the financial support from the National Natural Science Foundation of China (Grant Nos. 12274030, 11704075) and the National Key Research and Development Program of China (Grant No. 2022YFA1403400). Part of the experimental work was carried out in the Fudan Nanofabrication Lab.

## Author contributions

H.Y. and Y.L. initiated the project and conceived the experiments. Y.L. prepared the samples, performed the measurements and data analysis with assistance from J.L., S.H., B.Y., F.W., Q.X., C.S., C.W., Y.X., L.M., J.Z., and Y.M. J.M. and Y.L. provided the theoretical support. H.Y. and Y. L. cowrote the paper and all authors commented on the paper. H.Y. supervised the whole project.

## Competing interests

The authors declare no competing interests.
