## [Peer Review File · Nature Communications]

Reviewers' Comments:

Reviewer #1:

Remarks to the Author:

In this work, photocurrent is used to determine the photo-physics of excitons in 3 to 11 layer black phosphorus, focusing on the layer-dependent exciton polarizability (extracted using a quantum well model) and carrier screening effect under gating, with the observation that dark excitons are brightened under electrical gating. The major development of this paper is the identification of the exciton polarizability as a function of layer number. Otherwise, the evolution of the quasiparticle gap and exciton binding energy with layer number is something that has already been widely explored in a large number of published papers. Overall, the work in this paper appears well-done (though the theoretical analysis is somewhat lacking) and may be of interest to the 2D community in further characterizing excitons, but the results are pretty much consistent with previous results on black phosphorus and other 2D materials. The extraction of the exciton polarizability involves fitting results that are consistent with previous experimental results to a fairly simple quantum well model. Thus, I do not believe this paper meets the level of novelty and impact in typical Nature Communications papers, and I suggest this paper be resubmitted to a more specialized journal.

I also have some additional suggestions for revision.

1. The study of excitons in encapsulated 3 to 11 layer black phosphorus is somewhat limiting because, as seen in this paper (<https://www.nature.com/articles/nnano.2016.171>), the exciton binding energy in black phosphorus encapsulated by h-BN is considerably screened out by h-BN by the time one reaches four layers. Furthermore, the overall screening environment seen by excitons in few-layer BP and thicker BP is probably quite different, since h-BN has a much larger dielectric constant than BP, making the fundamental layer-dependent behavior difficult to extrapolate. Thus, it might be more informative to look at unencapsulated samples in vacuum and also try to go down to the monolayer limit. Perhaps I missed it, but the above paper should also be cited.
2. Following above, when considering the relative permittivity of h-BN, it is important to consider both the in-plane and out-of-plane permittivity, which differ considerably.
3. The carrier screening should be quite complicated, screening both the QP band gap and exciton binding energies. The authors account for this screening using a Thomas-fermi model, which I do not think is sufficient to capture these complicated, frequency dependent behaviors. This can be seen by Fig. 3, where the agreement between theory and experiment is quite poor.
4. Furthermore, it is not clear why a screened quantum well model is only needed above 8 layers. Surely, there is still carrier screening below 8 layers.

Reviewer #2:

Remarks to the Author:

In this work, Lei et al. measure the photocurrent in dual-gate black phosphorus (BP) field-effect transistor devices as functions of vertical electric field and material thickness, using a photocurrent spectroscopy technique. Their results demonstrate the field- and thickness-tunable excitonic responses. More importantly, by applying a vertical electric field and thus breaking the inversion symmetry, otherwise forbidden transitions are allowed and become stronger than the regular transitions. Despite the novelty, here are my questions and general suggestions the authors may consider.

1. Fig. 2a. The authors said this was 4-terminal measurement. Then why the measured quantity was source/drain current instead of conductance? Also, since this was 4-terminal measurement, ideally intrinsic material conductance was measured. But in the panel, I do not see bandgap modulation effect (e.g., consistent shift of the charge-neutrality conductance). As a result, it is not consistent with Fig. 2d.
2. The fitting of Fig. 3b is pretty unsatisfactory. The authors may consider focusing more on qualitative trend rather than pursuing quantitative agreement. Also, Fig. S9. What is the physics behind this fitting?

3. According to Fig. S14, $l=2$ subbands are shifting to higher energy? Does this contradict to the trend of E22 transition in Fig. 4b?

4. From my personal perspective, the novelties of this manuscript are mainly the application of the photocurrent spectroscopy technique to BP (Fig. 1) and the observation of dark excitons which are forbidden in the absence of a vertical electric field (Fig. 4). Fig. 2 and Fig. 3 are more or less known within the community. I would suggest the authors put more emphasis (both in terms of figures and text) on these novelties.

In summary, the manuscript presents an interesting study of the excitonic responses in BP under a vertical electric field. I would recommend the publication of the manuscript in Nature Communications provided that the authors consider the concerns at listed above.

Reviewer #3:

Remarks to the Author:

The manuscript entitled "Layer-dependent exciton polarizability and the brightening of dark excitons in few-layer black phosphorus" by Yuchen Lei et al. presents an interesting work on photocurrent generation in few-layer black phosphorus with a control of electric field. The authors measured exciton resonance energy as a function of electric fields and thickness where they quantify exciton polarizability for quantum confined stark effect in few-layer black phosphorus. The authors provide a physical model based on 1D quantum well under an electric field in addition to the electrostatic screening effect. In addition, the authors observe absorption resonances from symmetry-forbidden transitions in strong field regime. Although the manuscript reports interesting observation on excitons in vdW 2d semiconductors, but I have several questions and concerns on their analysis and viewpoints.

1. As shown in Fig.2d and Fig.3a, they observe the quadratic shift of exciton resonances under electric fields, which is expected in symmetric quantum well without a dipole moment. However, thickness dependence of exciton polarizability seems to show significant discrepancy to their model. They employ QW model below 8L while they employ screened QW model above 8L. The screened QW model incorporates dielectric constant contribution from thermally excited carriers. In Fig.3b, experimental data seems to show somehow linear-like dependence while both the QW model and the screened QW model predicts deviation from linear dependence. Also, 8L and 9L have nearly similar bandgap size according to their previous published work. I am wondering why they employ two different models for two thicknesses.

2. If they believe thermally excited carriers affect electric screening in their devices, can they provide temperature dependence of their measured polarizability? For Fig.4, they do have 10-layer device data at 77 K. I am wondering whether they observe different polarizability for two different temperatures due to the screening effect. This dependence will be helpful to support their screened QW model.

3. Can the author explain more detailed how their results provide physical picture on exciton evolution from 2D to 3D? In my opinion, their analysis is largely based on 2D QW. In this regard, would it be good to provide experimental data from significantly thicker sample like a near bulk which will correspond to 3D?

Response Letter

Reviewer #1 (Remarks to the Author):

In this work, photocurrent is used to determine the photo-physics of excitons in 3-to-11-layer black phosphorus, focusing on the layer-dependent exciton polarizability (extracted using a quantum well model) and carrier screening effect under gating, with the observation that dark excitons are brightened under electrical gating. The major development of this paper is the identification of the exciton polarizability as a function of layer number. Otherwise, the evolution of the quasiparticle gap and exciton binding energy with layer number is something that has already been widely explored in a large number of published papers.

Reply: We thank the reviewer for his/her careful reading and evaluation of our work. Apart from the layer dependent exciton polarizabilities, the index dependent behavior of excitons under gating and the brightening of dark excitons are also major developments of this paper. We agree with the referee that the evolution of the quasiparticle bandgap and exciton binding energy with layer number are not new. As a matter of fact, we never claimed or intended to claim those as our new findings in the manuscript.

Overall, the work in this paper appears well-done (though the theoretical analysis is somewhat lacking) and may be of interest to the 2D community in further characterizing excitons, but the results are pretty much consistent with previous results on black phosphorus and other 2D materials.

Reply: The decrease of the E_{11} exciton energy under an electrical field is indeed consistent with previous electrical gap behavior of black phosphorous (Nat Commun 8, 14474 (2017)), which was revealed by pure transport measurement. This scenario is also consistent with excitons in other 2D semiconductors. However, the systematic layer-dependence, the higher-index exciton response and the brightening of dark

excitons have not been investigated.

The extraction of the exciton polarizability involves fitting results that are consistent with previous experimental results to a fairly simple quantum well model. Thus, I do not believe this paper meets the level of novelty and impact in typical Nature Communications papers, and I suggest this paper be resubmitted to a more specialized journal.

Reply: The simple quantum well model does successfully predict the quadratic dependence of the exciton energy. However, it gives a L^4 scaling (L is the thickness or layer number) for the exciton polarizability. Our study has obtained the layer dependence of the exciton polarizability experimentally and we find that this deviates apparently from the L^4 scaling. So the simple quantum well model doesn't work well and we have to invoke other mechanisms.

In addition, the novelty of our paper also lies in the clear observation of higher index transitions under the electric field. Benefiting from the spectrum-resolved photocurrent measurement, the higher index excitons are clearly observed along with their index dependent behaviors under an electric field. This distinguishes our paper from previous studies, where only the quasiparticle band gap is the main focus. Even more interestingly, we also observe brightened forbidden transitions. Taking advantage of the good signal to noise ratio provided by the measurement method, we performed more studies on the dark excitons in the revision process. Now the peak positions of the first two dark excitons for several samples with various thickness are extracted experimentally for the first time, which leads to the determination of the interlayer coupling parameters for conduction and valences bands respectively.

In summary, our experimental results are partially as expected, but they are still full of surprises. We believe the manuscript meets the criteria of Nature Communications.

I also have some additional suggestions for revision.

1. The study of excitons in encapsulated 3-to-11-layer black phosphorus is somewhat limiting because, as seen in this paper

(<https://www.nature.com/articles/nnano.2016.171>), the exciton binding energy in black phosphorus encapsulated by h-BN is considerably screened out by h-BN by the time one reaches four layers. Furthermore, the overall screening environment seen by excitons in few-layer BP and thicker BP is probably quite different, since h-BN has a much larger dielectric constant than BP, making the fundamental layer-dependent behavior difficult to extrapolate. Thus, it might be more informative to look at unencapsulated samples in vacuum and also try to go down to the monolayer limit. Perhaps I missed it, but the above paper should also be cited.

Reply: This is a very insightful comment! The paper mentioned by the reviewer is a pioneering work on BP optical properties. We are sorry for missing it in the manuscript. Now we have cited it in the revision.

It's certainly a great idea to look at unencapsulated samples in vacuum, but due to the lack of feasible dual-gating scheme, we are not able to conduct such experiments in a short period. In fact, the encapsulated BP samples are more common in real experiments, so our findings are even more valuable to other experimentalists.

As for the dielectric screening effect on the exciton binding energy by encapsulating hBN, we'd like to stress the following:

First, we are looking at the external electrical field on the exciton peak shift, which originates from the shrinkage of the band-to-band transition energy and the reduction of the exciton binding energy under an E-field. Those two contributions have opposite effect on the exciton transition energy. However, literatures point out that the band-to-band energy shift largely dominates over the change of the exciton binding energy, consistent with the observed redshift of exciton peaks. Therefore, the common practice is to calculate the energy levels in a quantum well under the E-field and obtain the Stark shift, without invoking the reduction of the exciton binding energy. We adopted this picture as well in our analysis. As a result, the hBN screening effect on the binding energy for different thickness BP samples doesn't play a significant role in our experiment, since we are basically monitoring the change of the band-to-band transition energy under an E-field.

Second, from an experimental point of view, even though the exciton binding energy

for the encapsulated samples might be largely reduced and even become zero for samples thicker than 4L, as suggested by the mentioned paper, we can still observe quite sharp exciton resonance peaks for all samples up to 20L. This enables us to accurately track the exciton energy shift under an E-field and perform the layer dependence for the exciton polarizability. So we don't have to eliminate the encapsulating hBN to observe prominent excitons. Now our experimental scheme is to look at the energy shift of different thickness BP encapsulated by hBN under an E-field. Isn't it a clean experiment already?

2. Following above, when considering the relative permittivity of h-BN, it is important to consider both the in-plane and out-of-plane permittivity, which differ considerably.

Reply: This is also a stimulating question. If one examines the screening effect of hBN on the exciton binding energy, because the electrical field of excitons have both in-plane and out-of-plane components, one has to use the full version of hBN permittivity. However, for our case, the only relevant situation is the screening effect on our external vertical field, so we use the out-of-plane permittivity to calculate the exerted electrical field on BP from the applied voltage.

3. The carrier screening should be quite complicated, screening both the QP band gap and exciton binding energies. The authors account for this screening using a Thomas-fermi model, which I do not think is sufficient to capture these complicated, frequency dependent behaviors. This can be seen by Fig. 3, where the agreement between theory and experiment is quite poor.

Reply: The referee is right. Thomas-fermi model is certainly inadequate to account for the QP band gap and exciton binding energy renormalizations by free carrier screening. Nevertheless, it's not so relevant in our study. By using the simple model, we have already captured the main factor on the electric field effect.

To illustrate this, let's compare two cases. The first is an intrinsic BP without any free carrier within (we call it BP1 in the subsequent discussion), and the second is a BP (BP2) with some free carriers. Both of them have the same thickness.

First, without applying gate field, we can get E_{11} transitions in both samples. The peak positions might be different for BP1 and BP2, because the aforementioned renormalizations of QP band gap and exciton binding energy due to the free carrier screening in BP2. However, the peak positions may not be so different, since the QP band gap and exciton binding energy renormalizations have opposite effect on the E_{11} exciton transition energy and they are largely cancelled out, as demonstrated in TMDCs (*Nat Mater* 13, 1091-1095 (2014)). This scenario has the same physical origin as the negligible exciton peak position shift after BP is encapsulated by hBN, as demonstrated by our measurement. We compared the exciton peak positions of BP on PDMS substrate and encapsulated by hBN. As shown in Fig. R1, there is almost no shift, even though the screening from both top and bottom hBN is much stronger than the PDMS substrate. The conclusion is also corroborated by theoretical calculations from Louie's group (*Nano Lett* 17, 4706-4712 (2017)). In short, the exciton peak position depends on the screening condition very weakly.

Now we apply a vertical E-field to both samples. The major difference between BP1 and BP2 is that the local field in BP2 is weaker, because free carriers there screen the applied field. This makes the exciton positions of BP1 and BP2 different, even though the applied external fields are the same. Thomas-fermi model describes how the carriers redistribute and screen the field, which is responsible for the different exciton peak positions in BP1 and BP2. Now, the referee may argue, the gate field in BP2 redistributes the free carriers, which can change their screening capability on QP band gap and exciton binding energy, hence the measured peak position. However, as argued above, the peak position depends on the screening condition very weakly and we can safely neglect the external field induced change on the free carrier screening.

In summary, in our interpretation of the free carrier screening effect on the exciton polarizability in few-layer BP, Thomas-fermi model captures the main physical mechanism.

Fig. R1 The characterization of a 3-layer BP sample. a) The extinction spectrum of the sample on PDMS measured before FET fabrication. b) The normalized photocurrent spectrum measured after FET assembly, with BP encapsulated by hBN. The positions of E_{11} exciton is pointed out by red arrows and the exact energies are marked. The change of the peak position is only 5 meV.

4. Furthermore, it is not clear why a screened quantum well model is only needed above 8 layers. Surely, there is still carrier screening below 8 layers.

Reply: This is a very nice question. The screening depends on the free carrier density, which comes from the thermal excitation since we examine the Stark effect at the charge-neutral point, avoiding doping from the gate. The thermal carrier density sensitively depends on the band gap. Therefore, when BP layer number decreases, which drastically increases the band gap, the carrier density decreases rapidly. That's why we didn't adopt the screened quantum well model for very thin samples.

In more detail, using a non-linear Thomas-Fermi model, the screening effect caused by thermally excited carriers is brought in to correct the field to a lower level, which leads to final states with minimal energies. But the calculation process does not consider whether the density of free carriers could satisfy its need or not. Insufficient amount of the thermally excited carriers would lead to the overestimation of the screening effect by the model. To avoid the scenario, the densities of thermally excited carriers under electric field are calculated using a simple model shown in Fig. R2. The comparison between the densities of thermally excited carriers and densities required for the model

to work is shown in Table R1. As shown in Tab. R1, a 7-layer BP could only provide 20% of the density Toms-Fermi model needs. For BP samples thinner than 7-layer, the densities of thermal excited carrier are negligible compared to the densities Toms-Fermi model needs. As a result, the existence of free carrier screening effect is ignored in these samples. For BP samples thicker than 8-layer, the calculated densities become adequate for Thomas-Fermi model, so free carrier screening is considered in these samples. Indeed, changing the model from 8-layer BP is a little abrupt, so we modified Figure 3b according to the reviewer's comments, showing all the calculated results of both models. As we can see from Fig. R3, the 8- to 9-layer BP samples are located in the intermediate area of the two models.

Fig. R2 The schematic diagram of the thermal excitation model, with the applied external field along z direction. As the field and thickness are known, we could estimate the overall thermally excited carrier densities under different circumstances.

Fig. R3 The modified Fig. 3b which introduces the experimental polarizabilities and that calculated by the QW and screened QW models.

Layer number	Charge densities Thomas-Fermi model needs (cm ⁻²)	Charge densities of thermal excitation at 300 K (cm ⁻²)	Ratio of densities could provide versus needed
3	7.85×10^{11}	6.94×10^6	8.83×10^{-6}
4	1.13×10^{12}	4.82×10^8	4.29×10^{-4}
5	1.48×10^{12}	1.16×10^{10}	7.85×10^{-3}
6	1.79×10^{12}	9.28×10^{10}	5.17×10^{-2}
7	2.09×10^{12}	4.29×10^{11}	0.215
8	2.34×10^{12}	1.13×10^{12}	0.482
9	2.58×10^{12}	2.10×10^{12}	0.812
10	2.79×10^{12}	2.98×10^{12}	1.07
11	2.96×10^{12}	3.55×10^{12}	1.20

Tab. R1 The calculated thermally excited carrier densities of few-layer BP systems.

Reviewer #2 (Remarks to the Author):

In this work, Lei et al. measure the photocurrent in dual-gate black phosphorus (BP) field-effect transistor devices as functions of vertical electric field and material thickness, using a photocurrent spectroscopy technique. Their results demonstrate the field- and thickness-tunable excitonic responses. More importantly, by applying a vertical electric field and thus breaking the inversion symmetry, otherwise forbidden transitions are allowed and become stronger than the regular transitions. Despite the novelty, here are my questions and general suggestions the authors may consider.

Reply: We thank the referee for his/her encouraging assessment on our work.

1. Fig. 2a. The authors said this was 4-terminal measurement. Then why the measured quantity was source/drain current instead of conductance? Also, since this was 4-terminal measurement, ideally intrinsic material conductance was measured. But in the panel, I do not see bandgap modulation effect (e.g., consistent shift of the charge-neutrality conductance). As a result, it is not consistent with Fig. 2d.

Reply: We thank the reviewer for his/her careful reading and evaluation of our work. The suggestion is very reasonable and professional. The source/drain current is directly obtained from our 4-terminal measurement. Since the source-drain voltage is fixed to 0.2V in the measurement process, the source/drain current is proportional to the conductance. We have made modifications in Fig. 2a, replacing source/drain current by conductance.

The ratio of intrinsic black phosphorus conductance under different electric fields is proportional to an exponential term related to the energy gap and temperature (Nat Commun 8, 14474 (2017)):

$$N = \text{const} \cdot e^{-\frac{E_c - E_v}{2k_B T}} = \text{const} \cdot e^{-\frac{E_g}{2k_B T}}. \quad (\text{R2})$$

Let's make an estimate for the 6-layer sample. According to the photocurrent spectra, the optical gap decreases from 0.5517 eV to 0.5363 eV, dropping 0.0115 eV under a top gate voltage $V_{tg} = -6$ V. According to Eq. R2, the ratio of the resistance is equal

to $e^{-\frac{\Delta E_g}{2k_B T}} = 1.33$. As a result, the source-drain current at charge neutral point should gain 8×10^{-10} A, as predicted by the theory, which is beyond our measurement sensitivity. For thicker samples, the modulation of the bandgap becomes more significant, which leads to a larger change of the conductance. Figure R4 displays the 4-terminal measurement results of a 13-layer BP. It can be estimated that the band gap reduces 0.0998 eV under a $F = D/\epsilon_0 = 0.60$ V/nm displacement field ($U_{tg} = 10$ V). As we can see, now the result is fully consistent with the trend in previous study (*Nat Commun* 8, 14474 (2017)).

In summary, the modulation effect on the minimal conductance can be clearly observed in 4-terminal measurements of thicker samples and is limited by the sensitivity of the apparatus for thinner BP samples.

Fig. R4 The 4-terminal measurement results of a 13-layer BP sample. The U_{tg} interval between neighboring curves is 2 V.

2. *The fitting of Fig. 3b is pretty unsatisfactory. The authors may consider focusing more on qualitative trend rather than pursuing quantitative agreement. Also, Fig. S9. What is the physics behind this fitting?*

Reply: We thank the referee for the good suggestion.

Former studies based on the second-order perturbation theory have concluded that the second-order energy shift of subbands under a vertical field is promotional to the fourth power of the well width (*Phys Rev B Condens Matter* 33, 8385-8389 (1986)):

$$\Delta E_l^{(2)} = C_l F^2 L^4, \quad (\text{R3})$$

where $F = D/\varepsilon_0$ is the electric field, C_l is a constant related to the transition index l , and L is the width of the QW. Compared with Eq. R3, our results exhibit a more moderate relation between ΔE and L . The role played by the fitting in Fig. S9 is an illustration that our case deviates greatly from the second-order perturbation theory. As a result, modifications are needed.

3. According to Fig. S14, $l = 2$ subbands are shifting to higher energy? Does this contradict to the trend of E_{22} transition in Fig. 4b?

Reply: Figure R5 shows the experimental and calculated shifting of the E_{22} peaks of a larger range. It is obvious that the shifting trend of the numerical calculation is not monotonic in the overall range. As we can see from Fig. R5, our data fit the calculated trend for the first few points, but contradict with it as the field increases. Two possible reasons could account for the phenomenon above. First, unlike the E_{11} exciton peaks, the E_{22} peaks are broader, which leads to a larger uncertainty in extracting the peak positions. Second, $\Delta n = 2$ transitions may occur under considerable field (in our case, E_{13} transition). With the transition energy very close to the E_{22} peak (*ACS Nano* 17, 6073-6080 (2023)), the two peaks may overlap into a broader one, which may have an impact on the position extraction as well. Nevertheless, the slower shift of E_{22} than that of E_{11} excitons is evident in our data, which is consistent with theory.

Fig. R5 The experimental and numerically calculated shifts of E_{22} exciton in a 10-

layer BP.

4. From my personal perspective, the novelties of this manuscript are mainly the application of the photocurrent spectroscopy technique to BP (Fig. 1) and the observation of dark excitons which are forbidden in the absence of a vertical electric field (Fig. 4). Fig. 2 and Fig. 3 are more or less known within the community. I would suggest the authors put more emphasis (both in terms of figures and text) on these novelties.

Reply: We thank the referee for the insightful evaluation of our work and the good suggestion. Following the referee's suggestion, additional experiments and analysis on the brightening of dark excitons have been carried out. The photocurrent spectra of 8-layer and 9-layer BP under room and cryogenic temperatures have been systematically measured, which not only further confirm the brightening of dark excitons, but also allow us to extract the exact splitting of the conduction and valence bands respectively, as displayed in Fig. R6. Benefited from the observation of the two dark excitons (E_{21} and E_{12}), the respective interlayer coupling for the conduction and valence bands (described as γ_c and γ_v , *ACS Nano* 17, 6073-6080 (2023)) have been obtained by fitting the splitting of bands extracted from 8- to 10-layer data shown in the paper and supplementary information. Now we have a new Fig. 4c and have added corresponding discussions. It's worth noting that, without the emergence of forbidden transitions in the spectra, γ_c and γ_v cannot be determined separately, though the combined value $\gamma_c - \gamma_v$ is available.

Fig. R6 The newly added Fig. 4c in the main text, which displays the splitting of the first two conduction and valence bands extracted from the photocurrent spectra displayed in Fig. 4a, SI Fig. 18 and SI Fig. 19. The curves are the fitting results of the data points, with γ_c and γ_v as fitting parameters.

In summary, the manuscript presents an interesting study of the excitonic responses in BP under a vertical electric field. I would recommend the publication of the manuscript in Nature Communications provided that the authors consider the concerns at listed above.

Reply: We thank the referee again for his/her positive evaluation of our work.

Reviewer #3 (Remarks to the Author):

The manuscript entitled “Layer-dependent exciton polarizability and the brightening of dark excitons in few-layer black phosphorus” by Yuchen Lei et al. presents an interesting work on photocurrent generation in few-layer black phosphorus with a control of electric field. The authors measured exciton resonance energy as a function of electric fields and thickness where they quantify exciton polarizability for quantum confined stark effect in few-layer black phosphorus. The authors provide a physical model based on 1D quantum well under an electric field in addition to the electrostatic screening effect. In addition, the authors observe absorption resonances from symmetry-forbidden transitions in strong field regime. Although the manuscript reports interesting observation on excitons in vdW 2d semiconductors, but I have several questions and concerns on their analysis and viewpoints.

Reply: We thank the referee for his/her careful reading and evaluation of our work

1. As shown in Fig.2d and Fig.3a, they observe the quadratic shift of exciton resonances under electric fields, which is expected in symmetric quantum well without a dipole moment. However, thickness dependence of exciton polarizability seems to show significant discrepancy to their model. They employ QW model below 8L while they employ screened QW model above 8L. The screened QW model incorporates dielectric constant contribution from thermally excited carriers. In Fig.3b, experimental data seems to show somehow linear-like dependence while both the QW model and the screened QW model predicts deviation from linear dependence. Also, 8L and 9L have nearly similar bandgap size according to their previous published work. I am wondering why they employ two different models for two thicknesses.

Reply: Compared to the real sample, QW is a fairly simple model. Moreover, the uniformity of the electric field in the well is an assumption we made to simplify the calculation, because the redistribution of the thermal carriers certainly causes non-uniformity to the field. Therefore, the experimental polarizabilities do deviate from the calculated polarizabilities. But the numerical difference between them is not very

significant after taking the free carrier screening effect into account for the relatively thicker samples. Qualitatively, the layer dependent trends of the experimental and theoretical E_{11} polarizabilities are largely consistent. Considering all the above factors, we believe that our theory can describe our experiments to a certain extent.

As for why we used two models for 8L and 9L respectively, we did some calculations to justify. The screening effect caused by thermally excited carriers is brought in to correct the field to a lower level using a non-linear Thomas-Fermi model. The model leads us to final states with minimal energies, but does not consider whether the density of free carriers could satisfy its need or not. To verify this point, the densities of thermally excited carriers under electric field are calculated using a simple model, as shown in Fig. R7. The comparison between the densities of thermally excited carriers and densities required for the model to work is shown in the Table R2.

The densities of thermally excited carriers are negligible compared to the demand of the model in relatively thin BP. As a result, the screening effect could be ignored in 6-layer and thinner BP. For BP thicker than 9-layer, the calculated densities of thermally excited carriers could completely satisfy the demand of our screened model. Therefore, for these BP systems, the screened QW model was adopted. As for the intermediate range, the thermally excited carriers are not adequate for the model but cannot be ignored either. Therefore, in our manuscript, the sample thickness boundary of the two models was set at 9-layer. The reviewer's question is very reasonable, it does seem abrupt with the model changed from 8-layer to 9-layer. Now modification has been made in our revised Fig. 3b in the main text to eliminate such a sudden change. As shown in Fig. R8, we now display all the results of QW model and screened QW model calculations for each thickness. Admittedly, neither of the models is perfect for all the measured samples, with the QW model only working well for the thin ones and the screened QW model working better for the thicker ones. But as an experimental study which measures the polarizability, the experimental data themselves are the most important part. As suggested by reviewer#2, a qualitative agreement on the layer-dependent trend might be sufficient.

Fig. R7 The schematic diagram of the thermal excitation model, with the applied external field along z direction. As the field and thickness are known, we could estimate the overall thermally excited carrier densities under different circumstances.

Fig. R8 The modified Fig. 3b which introduces the experimental polarizabilities and that calculated by the QW and screened QW models.

Layer number	Charge densities Thomas-Fermi model needs (cm^{-2})	Charge densities of thermal excitation at 300 K (cm^{-2})	Ratio of densities could provide versus needed
3	7.85×10^{11}	6.94×10^6	8.83×10^{-6}
4	1.13×10^{12}	4.82×10^8	4.29×10^{-4}
5	1.48×10^{12}	1.16×10^{10}	7.85×10^{-3}
6	1.79×10^{12}	9.28×10^{10}	5.17×10^{-2}

7	2.09×10^{12}	4.29×10^{11}	0.215
8	2.34×10^{12}	1.13×10^{12}	0.482
9	2.58×10^{12}	2.10×10^{12}	0.812
10	2.79×10^{12}	2.98×10^{12}	1.07
11	2.96×10^{12}	3.55×10^{12}	1.20

Tab. R2 The calculated thermally excited carrier densities of multi-layer BP systems.

2. If they believe thermally excited carriers affect electric screening in their devices, can they provide temperature dependence of their measured polarizability? For Fig.4, they do have 10-layer device data at 77 K. I am wondering whether they observe different polarizability for two different temperatures due to the screening effect. This dependence will be helpful to support their screened QW model.

Reply: This is a very good suggestion.

The density of thermally excited carriers is given by the equation below:

$$N = \int_{E_c}^{\infty} g(E)F(E) dE, \quad (\text{R4})$$

where E_c is determined by both the position of the BP calculated in z direction and E_g , $g(E)$ is the density of states and $F(E)$ is the Fermi-Dirac distribution:

$$F(E) = \frac{1}{1 + e^{\frac{E-E_F}{k_B T}}}. \quad (\text{R5})$$

Equation R4 implies that the densities of thermally excited carriers are determined by both the band gap and the temperature. As the temperature drops, the band gap of BP also decreases (*Phys Rev Lett* 125, 156802 (2020)). Take the 9L BP system for clarify. The energy of E_{11} peak decreases from 0.468 eV at room temperature to 0.452 eV at 77.5 K. As a result, lowering temperature simultaneously alters $F(E)$ and E_c , making it difficult to simply size up the changes in densities. Numerical calculation was carried out to get detailed information, and the results are displayed in Table R3 under a universal field $F = D/\epsilon_0 = 0.8$ V/nm. The 8- to 10-layer BP are aimed because the densities of thermal carriers are in the transition regime from fully inadequate to fully

adequate to the Thomas-Fermi model, thus the results under 77.5 K are expected to vary from that under 300 K.

As shown in Table R3, the calculated ratios reveal little difference for 8- to 10-layer BP under room and cryogenic temperature. This indicates that the impact of temperature is nearly compensated by the change of the band gap.

Layer number	Charge densities Thomas-Fermi model needs (cm^{-2})	Charge densities thermal excitation provide at 300 K (cm^{-2})	Ratio of densities at 300 K	Charge densities thermal excitation provides at 77.5 K	Ratio of densities at 77.5 K
8	2.34×10^{12}	1.13×10^{12}	0.482	9.19×10^{11}	0.393
9	2.58×10^{12}	2.10×10^{12}	0.812	2.05×10^{12}	0.792
10	2.79×10^{12}	2.98×10^{12}	1.07	3.08×10^{12}	1.11

Table R3 The calculated thermally excited charge densities of 8-to-10-layer BP at room and cryogenic temperatures.

To verify the reliability, we performed further cryogenic experiment and present experimental results of photocurrent spectra of 8-layer and 9-layer BP measured at 77.5 K in revised SI Fig. 18 and SI Fig. 19, as suggested by the reviewer. Their polarizabilities are extracted and shown in Table R4, with the 10-layer experimental result extracted from Fig. 4a. It could be concluded from Table R4 that the difference between room temperature polarizabilities and cryogenic temperature ones is small, within the error bars. This is consistent with our thermal excitation calculations, proving the influence of temperature is nearly canceled out by the change of the band gap. Although we observed little difference between polarizabilities for two different temperatures, the results can not disapprove our model. So we still believe it's reasonable to attribute the screening effect to thermal carriers in relatively thick samples.

Layer number	E_{11} polarizabilities α at 300 K ($\text{eV} \cdot \text{nm}^2/\text{V}^2$)	E_{11} polarizabilities α at 77.5 K ($\text{eV} \cdot \text{nm}^2/\text{V}^2$)
8	-0.086 ± 0.004	-0.083 ± 0.003
9	-0.097 ± 0.002	-0.103 ± 0.003
10	-0.099 ± 0.004	-0.105 ± 0.008

Table R4 The polarizabilities of 8-to-10-layer BP extracted from experimental data measured at room and cryogenic temperatures.

3. *Can the author explain more detailed how their results provide physical picture on exciton evolution from 2D to 3D? In my opinion, their analysis is largely based on 2D QW. In this regard, would it be good to provide experimental data from significantly thicker sample like a near bulk which will correspond to 3D?*

Reply: This is a very nice point. Our results provide physical pictures on exciton evolution from 2D to 3D from two basic parameters.

The first parameter is the exciton polarizability. Since the space in the out-of-plane direction increases when the sample thickness increases, it's easier to separate the electron and hole in an exciton to induce a dipole, hence gives larger polarizability. This is exactly what we observed when the thickness increases from 3- to 11-layer. To some extent, our study certainly provides information on the polarizability evolution from 2D to 3D. Of course, the reviewer's suggestion is very insightful. It's better to determine the polarizability for a significantly thicker sample (safely corresponds to 3D case) to have a more complete picture. We have followed the suggestion and actively pursued in this direction, and performed measurements for samples up to 20-layer. Unfortunately, due to the rapid decrease of the exciton oscillator strength under the field (the 2nd parameter discussed below), we have difficulty to determine the polarizability. The second parameter is the exciton oscillator strength. The major advantage of the quantum confined Stark effect (QCSE) is the robustness of excitons under the field. In our study, we gradually eased the confinement by increasing the sample thickness. We

observed that the oscillator strength decreases under the field more rapidly for relatively thick samples, particularly for the newly measured 13-, 18-, and 20-layer samples. This is fully consistent with the evolution from QCSE in quasi-2D to the regular Stark effect in 3D. The newly measured samples are detailed below and have been included in the revised paper.

Figure R9 displays the photocurrent spectrum of 13L BP sample. Before we can see some distinguishable shift of E_{11} exciton, the exciton resonance feature almost disappears. Such scenario is even more prominent for the 18- and 20-Layer samples. As seen in Fig. R10 and Fig. R11, the E_{11} peaks tend to disappear at even smaller electric field ($F = D/\epsilon_0 = 0.12$ V/nm and $F = 0.06$ V/nm for 18L and 20L BP samples respectively). This indicates that from 2D to 3D, the exciton becomes much less robust under the field.

Though we believe that we have provided valuable information on the exciton dimensional crossover from 2D to 3D, given that we don't get information on bulk samples, we have made our claim weaker in the revised manuscript. In the abstract, we have replaced "Our study not only sheds light on the exciton evolution **from 2D to 3D**, ..." with "Our study not only sheds light on the exciton evolution **with sample thickness**, ...".

Fig. R9 The photocurrent spectrum of a 13-layer BP measured at room temperature.

Fig. R10 The photocurrent spectrum of an 18-layer BP measured at room temperature.

Fig. R11 The photocurrent spectrum of a 20-layer BP measured at room temperature.

Reviewers' Comments:

Reviewer #1:

Remarks to the Author:

I thank the authors for addressing my comments. Based on their response and the revised manuscript, the main developments in this work are the extraction of the exciton polarizabilities as a function of layer number, and the observation of the dipole-forbidden E12 and E21 excitons for the first time. With the additional emphasis on the dark excitons, the novelty of this paper becomes much clearer, and it seems appropriate for Nature Communications. I have a few additional comments:

Since we are discussing excitons, it may be clearer to refer to the optical bandgap throughout the paper rather than just the band gap.

I am still not fully convinced by the quantum well and screened quantum well models and am not sure they add much to the discussion. If you compare the QW against experiment, the only way in which the model agrees with experiment is the increase in polarizability with layer number. The curvature is very different, and in fact, the experimental data looks like it has a number of step-like features, which don't really agree with either model. It could be that the model is missing some interesting physics that is happening between 4 and 6 layers. It would be helpful if some error bars could be included for the experimental data.

The claim about robustness of the exciton due to confinement is also tricky. Due to a combination of confinement and the linear dichroism in black phosphorus, the optical absorption will still have sharp peaks up to about 4 layers, even in the absence of electron-hole interactions. This may be why the oscillator strength is large even under a large field. I think this ambiguity does not detract from the experimental results, since the robustness of the large oscillator strength may be technologically useful, but it may be helpful to qualify the statement about the robustness of the "exciton" oscillator strength.

Reviewer #2:

Remarks to the Author:

The authors have addressed my comments well. I recommend publication of the manuscript in Nature Communications.

Reviewer #3:

Remarks to the Author:

Yuchen Lei et. al. has provided additional experimental data along with elaborated discussion which are indeed helpful to address previously raised questions. Although some of the questions are addressed, I have still a concern on their screen QW model. After carefully reading the revised main text and supplementary information, I receive an impression that the authors claim their experimental results can be well supported by the screened model. The line 200 in the revised manuscript, referring to Fig.3a, says "The screened QW model agrees reasonably well with experimental results". I agree with their claim that qualitatively the polarizability can be decreased due to the screening effect. However, as far as I understood, the way the author obtained the polarizability was from fitting based on the phenomenological quadratic model with arbitrary coefficients. If they believe their model with the screening effect provides good explanation, they should show good agreement between theory and experiments in Fig.3b where the thickness dependence of the polarizability has to be correctly predicted.

I agree with their explanation in the rebuttal that matching with a good quantitative agreement can be challenging at the moment. There can be additional effect such as non-uniformity of E-field effect and so on as the authors suggests. If so, in my opinion, they should tone down their claim, and leave a possibility for the future study.

Other than the concern above, I find their manuscript has been much more improved so I can recommend the publication in Nature Communications.

RESPONSE TO REVIEWERS' COMMENTS

Reviewer #1 (Remarks to the Author):

I thank the authors for addressing my comments. Based on their response and the revised manuscript, the main developments in this work are the extraction of the exciton polarizabilities as a function of layer number, and the observation of the dipole-forbidden E_{12} and E_{21} excitons for the first time. With the additional emphasis on the dark excitons, the novelty of this paper becomes much clearer, and it seems appropriate for Nature Communications. I have a few additional comments:

Reply: We thank the reviewer for his/her encouraging comments.

Since we are discussing excitons, it may be clearer to refer to the optical bandgap throughout the paper rather than just the band gap.

Reply: This is a very insightful comment. We now have replaced “excitons associated with the bandgap” with “optical bandgap” or “resonances associated with the optical bandgap” in the revised manuscript when referring to the E_{11} excitons.

However, when we discuss the calculation of the thermally excited carriers in the paper, we still keep the terminology “bandgap”, since that’s what we really mean.

I am still not fully convinced by the quantum well and screened quantum well models and am not sure they add much to the discussion. If you compare the QW against experiment, the only way in which the model agrees with experiment is the increase in polarizability with layer number. The curvature is very different, and in fact, the experimental data looks like it has a number of step-like features, which don't really agree with either model. It could be that the model is missing some interesting physics that is happening between 4 and 6 layers. It would be helpful if some error bars could be included for the experimental data.

Reply: We thank the reviewer for the nice suggestion. Error bars of the experimental

polarizabilities have been added in the revised Fig. 3b, as also shown in Fig. R1 below. Admittedly, neither model fits the overall experimental results perfectly. As for the QW model, the discrepancy may come from two sources.

The first is that the QW model is too idealized to account for the potential. As the calculated BP system becomes thinner, the influence of the puckered structure and the inhomogeneous atomic interaction gets more and more prominent, distorting the potential inside the well. As a result, the potential condition is non-uniform in the QW and is complicated near the edge of the QW. The difference between the real sample and the standard infinite square well leads to the deviation between the experiment and calculation in thin BP samples.

The second is the imprecise effective mass we adopted. Due to the lack of relevant research on the effective mass of thin films, we use the effective mass of the bulk rather than the thin film. As the sample gets thinner, the possibly more severe deviation of the effective mass results in the discrepancy between the experiment and calculation.

Although the QW model shows a quite different curvature against the experiment between 4 and 6 layers, the increasing trend happens to coincide. Moreover, the increasing of the polarizabilities is our main claim, which unravels the thickness dependent behavior of the E_{11} excitons. So, it is reasonable to claim that the QW model fits qualitatively with the experiment in relatively thin BP systems.

As for the possible interesting physics happening between 4 and 6 layers, we don't have firm experimental evidence for that. Meanwhile, there is no solid evidence for the possible step-like features in the layer-dependent polarizability either.

Fig. R1 The revised Fig. 3b in the manuscript. The error bars are included.

The claim about robustness of the exciton due to confinement is also tricky. Due to a combination of confinement and the linear dichroism in black phosphorus, the optical absorption will still have sharp peaks up to about 4 layers, even in the absence of electron-hole interactions. This may be why the oscillator strength is large even under a large field. I think this ambiguity does not detract from the experimental results, since the robustness of the large oscillator strength may be technologically useful, but it may be helpful to qualify the statement about the robustness of the “exciton” oscillator strength.

Reply: The referee brings up an interesting point that the in-plane anisotropy of black phosphorus may enhance the optical peaks in the experiment. We agree with the referee on this point. However, all our samples, regardless of the thickness, have the same in-plane anisotropy. So the only factor that matters is the thickness or the confinement. Therefore, we attribute the robustness of excitons in very thin samples under high field to the quantum confinement. This is very reasonable and consistent with quantum confined Stark effect in other quantum well systems as well.

Reviewer #3 (Remarks to the Author):

Yuchen Lei et. al. has provided additional experimental data along with elaborated discussion which are indeed helpful to address previously raised questions. Although some of the questions are addressed, I have still a concern on their screen QW model. After carefully reading the revised main text and supplementary information, I receive an impression that the authors claim their experimental results can be well supported by the screened model. The line 200 in the revised manuscript, referring to Fig.3a, says “The screened QW model agrees reasonably well with experimental results”. I agree with their claim that qualitatively the polarizability can be decreased due to the screening effect. However, as far as I understood, the way the author obtained the polarizability was from fitting based on the phenomenological quadratic model with arbitrary coefficients. If they believe their model with the screening effect provides good explanation, they should show good agreement between theory and experiments in Fig.3b where the thickness dependence of the polarizability has to be correctly predicted.

Reply: We thank the reviewer for his/her careful reading of our work. Compared with the real sample, QW model is too idealized. The differences in sample component, thickness and structure are all absorbed in the effective mass, which we adopted the bulk value for all samples, regardless of the thickness. Consequently, QW model lacks adequate consideration of these vital ingredients. Moreover, the screened QW model we recalculated does deviate from the actual ones because of the uniform E-field approximation we made to simplify the calculation. As a result, both the QW and the screened QW model fail to show perfect numerical agreement with the experiment. By stating that “The screened QW model agrees reasonably well with experiment results”, we meant to emphasize the increasing trends are consistent although the numerical results do deviate from each other. After reconsideration, we do feel that the claim we made is too strong. In the manuscript, we have replaced “The screened QW model agrees **reasonably** well with experimental results” with “The screened QW model agrees **better** with experimental results **than the QW model for**

relatively thick samples”.

I agree with their explanation in the rebuttal that matching with a good quantitative agreement can be challenging at the moment. There can be additional effect such as non-uniformity of E-field effect and so on as the authors suggests. If so, in my opinion, they should tone down their claim, and leave a possibility for the future study.

Reply: We thank the reviewer for his/her suggestion. The reviewer is right, the redistribution of the carriers distorts the uniformity of the E-field effect, making it hard to reach a good agreement with the experiment. We have replaced “The screened QW model agrees **reasonably** well with experimental results” with “The screened QW model agrees **better** with experimental results **than the QW model for relatively thick samples**” to make our claim more appropriate. More specifically, we have also added the sentence “**Admittedly, neither model can fit the overall data well enough, and more sophisticated theoretical work is required in the future**” when we conclude the description of Fig. 3b in the paper.

Other than the concern above, I find their manuscript has been much more improved so I can recommend the publication in Nature Communications.

Reply: We thank the reviewer for his/her positive evaluation of our work.